# Climate zones are a key component of the heterogeneous presentation of malaria and should be added as a malariometric for the planning of malaria elimination

Chander Prakash Yadav[1,2,3☯], Syed Shah Areeb Hussain[2,3☯], Rajit Mullick[2,3☯], Manju Rahi[2,3,4], Amit Sharma[5]*

**1** ICMR-National Institute of Cancer Prevention & Research (NICPR), Noida, UP, India, **2** ICMR-National Institute of Malaria Research (NIMR), New Delhi, India, **3** Academy of Scientific and Innovative Research (AcSIR), Ghaziabad, India, **4** Indian Council of Medical Research, New Delhi, India, **5** Molecular Medicine Group, International Centre for Genetic Engineering and Biotechnology, New Delhi, India

☯ These authors contributed equally to this work.
* amit.icgeb@gmail.com

**Data Availability Statement:** Malaria data before 2019 are available in the public domain (nvbdcpmohfw@nic.in); and data after 2019 are

## Abstract

Malaria is a climate-sensitive disease and different climatic conditions affect the propagation of malaria vectors thereby influencing malaria incidence. The present study was undertaken to delineate malaria distribution across different climate types and sub-types in India and assess its significance as a malariometric in the ongoing elimination activities. All Indian districts were classified into three major climatic zones (Tropical, Temperate, and others (Arid, Cold, and Polar) based on the Köppen-Geiger climate classification system. The Annual Parasite Incidence (API) of malaria was analyzed in these climatic zones using the Kruskal Wallis test, and a post hoc comparison was done using the rank-sum test with an adjusted p-value for the level of significance. Further logistic regression was used to investigate the association of these climatic zones with high malaria incidence (i.e., API>1). The majority of Indian districts fall in Temperate (N = 270/692 (39.0%)) and Tropical (N = 260/692 (37.6%)) regions, followed by Arid (N = 140/692 (20.2%)), Polar (N = 13/692 (1.9%)) and Cold (N = 9/692 (1.3%)) regions. Three climate zones: Arid, Polar, and Cold were similar in terms of malaria incidence over the years and thus were grouped into one. It was found that the tropical and temperate zones display a significantly higher burden of malaria as compared to others for the studied years (2016–2021). Future projections of climate suggest a significant expansion of tropical monsoon climate towards central and northern India, along with a growing footprint of tropical wet savannah climate in the northeast of India by 2100, which could increase the risk of malaria transmission in these regions. The heterogeneous climatic zones of India play an important role in malaria transmission and can be used as a malariometric for the stratification of districts destined for malaria elimination.

not yet available in the public domain but can be taken on request. Data requests can be made through the website (www.nvbdcp.gov.in/ E-Mail: nvbdcpmohfw@nic.in). Data for the 1km resolution maps of Koppen-Geiger climate classification for the present (1980-2016) as well as the future (2100) is available in the public domain from the website (http://www.gloh2o.org/koppen/) and is referenced in the study (doi:10.1038/sdata.2018.214).

**Funding:** The authors received no specific funding for this work.

**Competing interests:** The authors have declared that no competing interests exist.

## Introduction

Malaria is a serious public health concern with an estimated 241 million malaria infections and 627,000 fatalities in 2020 [1]. The South-East Asia Region (SEAR) of the World Health Organization accounted for ~ 2% of the global malaria burden in 2020, with 5 million cases and 9000 deaths, with India accounting for ~83% of the cases and ~82% of the fatalities [1]. Malaria is caused by the Plasmodium parasite (mainly *P. falciparum* and *P. vivax)* and transmitted by mosquitoes of the *Anopheline* genus. Mosquitoes are cold-blooded with aquatic larval stages. Climatic factors such as temperature and rainfall can therefore significantly influence their prevalence and distribution. Temperature also affects the rate of development of the plasmodium parasite in the mosquito vector. Consequently, malaria is considered a climate-sensitive disease and any change in climatic conditions may have a major effect on malaria epidemiology. Climatic factors affect transmissibility by altering the parasite growth rate, prevalence, and development in mosquitos. The climate of a region is classified either based on climatic controls factors that determine or control the climate (such as net radiation, air circulation etc.) or is based on observed climatic conditions (such as temperature, precipitation etc.) and their effect on other phenomena (vegetation, animals etc.). Bio-meteorological classification is a specialized system that classifies the climate based on its influence on living organisms. The Köppen Climate classification, first formulated by Wladimir Köppen in 1918 (1846–1940) the German climatologist, is one of the most widely used climate classification systems and it is based on the empirical relationship between climate and vegetation [2]. Koppen climate classification is an ecologically relevant system that is used extensively in the scientific literature. It uses seasonal patterns in temperature and precipitation to divide regions into five major climate types (A-Tropical, B-Arid, C-Temperate, D-Cold, and E-Polar) as well as into 30 different climate sub-groups within the different major climate types (Table A in S1 Text). The greatest advantage of the Koppen climate classification is that compared to other systems Koppen relies on the most basic climatic parameters i.e., temperature and precipitation which are relatively simpler to measure and observe. Furthermore, for the inclusion of factors related to evapotranspiration, due consideration is also given to the relationship between temperature and precipitation. The rate of evapotranspiration also affects the moisture requirement of plants, and therefore, the climate classes defined by Koppen have biological relevance [3]. This visible association between vegetation and climate types makes Koppen climate classification highly relevant to bio-geographic researchers.

Summer months in the temperate zone and humid lowlands in the tropical region are major contributors to malaria transmissibility [4,5]. Temperature ranges between 24˚C to 28˚C with a relative humidity of 55% to 80% is most suitable for both *Plasmodium vivax* and *Plasmodium falciparum* transmission [6]. Most tropical mosquito species digest blood meals in two to three days, depending on the temperature, while in colder, temperate regions, blood digestion in mosquito gut can take anywhere from seven to fourteen days [7]. In the tropics, eggs hatch in 2–3 days, but in countries with more temperate climates, they may take 7–14 days or longer to hatch [7]. Similarly, the pupal phase and larval development stages are significantly shorter in the tropics than in temperate regions. In this way, vector distributions and their ability to transmit the malaria parasite are largely dependent on the type of climate.

Being the seventh largest country in the world, India is climatically diverse with vastly differing landscapes across various parts of the country. The strikingly heterogenous climates in India play contrasting roles in supporting and propagating malaria vectors, and thereby directly influence the malaria incidence in the region. As India is set to eliminate malaria by 2030. We must take into cognizance the heterogeneity of malaria distribution in different zones of the country, and it is important to understand the variations in malaria transmission,

morbidity, and mortality at different temporal and spatial scales within the context of climate. This facet is further complicated due to climate change that is sweeping across the world and India is projected to bear a brunt of it in the coming decades. Here, we have probed the associations between the varied climate classes of India as per Koppen classification and the malaria burden in the country over six years. We show that an additional dimension of malaria metric i.e., climatic zone, is vital for understanding disease epidemiology and for assessing the progress of malaria elimination and monitoring the vulnerability for malaria transmission across India.

## Methodology

### Ethics approval

Ethical approval is not required for this study as it is an analysis of secondary data.

### Study data sets

To assess the pattern of malaria incidence over the past two decades in different climatic zones in India, high-resolution (1 km) raster maps of Koppen-Geiger Climate classification for the present (1980–2016) and future climates (2100) were obtained from freely available datasets [3]. The study classified the climate based on precipitation and temperature data from multiple different sources (including WorldClim V1; CHELSA V1.2 and CHPclim V1) to account for uncertainties, as well as applied correction for topographical effects to produce a highly accurate classification of climate at a fairly high level of resolution of 1 km. Future climate predictions were based on the high-emission scenario represented by the Representative Concentrative Pathway (RCP) 8.5. This is the worst-case scenario of climate change and was deemed more useful in the context of malaria elimination as it can help us better prepare for any future risk that may lead to malaria resurgence. Yearly data (2000–2021) on malaria incidence (API) at the district level were obtained from the National Centre for Vector-borne Diseases Control (NCVBDC), which is the central nodal agency responsible for collecting and recording data on six major vector-borne diseases including malaria in India—others being dengue, chikungunya, lymphatic filariasis, visceral leishmaniasis, and Japanese encephalitis.

### Description of Koppen-Geiger climate classes

The Koppen climate classification divides climates into six primary categories A, B, C, D, E, and H as per the detail given below.

1. **A: Tropical rain climates** correspond to the regions in which the mean temperature of the coldest month exceeds +18˚C, and the annual precipitation amount is higher than the aridity threshold defined for type B.

2. **B: Arid/Dry climates** represents regions where the annual mean precipitation is lower than the evapotranspiration rate estimated in terms of the temperature-precipitation index (aridity threshold). This climate type is generally characterized by relatively lower rainfall, which may be during summer, winter, or is undefined.

3. **C: Temperate rain climates** are regions where the mean temperature of the coldest month should be between –3˚C and +18˚C. The precipitation amounts must be higher than the aridity threshold.

4. **D: Cold/continental climates** represent regions where the mean temperature of the warmest month must be higher than 10˚C, and the coldest month temperature should be below −3˚C, and precipitation amounts exceed the aridity threshold.

5. **E: Polar/alpine climates** is defined according to the mean temperature of the warmest month that must be lower than 10˚C.

6. **F: Highland climates** are colder regions due to elevation. Temperature and precipitation characteristics are highly dependent on traits of adjacent zones and overall elevation—highland climates are more dependent on the altitude rather than the latitude.

There are several different subgroups within each of the major climate groups that represent the precipitation seasonality. These subgroups are denoted by different letters placed after the major climate type and represent the seasonal distribution of precipitation and additional temperature characteristics. For example, the addition of the symbol 'a' after the major climate type indicates a hot summer (average temperature > 22˚C), 'b' denotes warm summers (average temperature < 22˚C, but temperature of at least four warmest months > 10˚C), 'c' denotes cool summers (average temperatures below 22˚C), 'd' denotes very cold winter (temperature of coldest month < -38˚C), 'f' refers to an absence of dry season (precipitation of driest month > 60 mm), 'w' represents dry winters (precipitation of driest month in winter half < 1/10th precipitation of wettest month of summer half), 's' represent dry summers (precipitation in the driest month of summer half < 30 mm), 'h' denotes an annual average temperature above 18˚C, 'k' denotes an annual average temperature below 18˚C and 'm' refers to monsoon. Besides these, capital letters are also used where 'S' denotes a Steppe type of climate, 'W' denotes a desert type climate, 'T' denotes a Tundra climate and 'F' denotes perennial frost. A complete description of the Koppen classification with its all subtype is given in supporting information (Table A in S1 Text).

**Annual Parasite Incidence (API).**   The number of confirmed new malaria cases expressed per 1,000 individuals at risk under surveillance [8] and can be written as

$$\text{API} = \frac{\text{Total positive}}{Total\ population} * 1000$$

API is the main criterion for classifying the districts into different categories. National Centre for Vector-Borne Diseases (NVBDC) uses API to define malaria transmission and classifies all Indian districts for malaria intervention into different categories as given below.

- Category 1 –Prevention of re-establishment phase (API = 0 or zero malaria cases),

- Category 2 –Elimination phase (State malaria API<1, and all districts having API<1),

- Category 2 –Pre-elimination phase (State malaria API<1, but some districts have API>1),

- Category 3 –Intensified control phase (State malaria API ≥1) [9].

## Statistical analysis

To probe the association between climatic zones and malaria burden, the annual parasite incidence (API) of Indian districts was compared across all major climate classifications using the Kruskal Wallis test, multi-level growth models, and Box / whisker plots. As our preliminary analysis suggested that the three climate zones viz. Arid (B), Cold (D), and Polar (E) were similar in terms of malaria burden, we combined these into one. Hence, the five major climate zones were collapsed into three: Tropical, Temperate, and Others (that includes Arid, Cold,

and Polar). Thereafter, malaria APIs were compared over three climate zones using the Kruskal Wallis test, a non-parametric counterpart of analysis of variance (ANOVA), and then post hoc analysis was done for pairwise comparison between groups using the rank sum test by adjusting the p-value. The district APIs were categorized into two (API <1 and API ≥1) and compared among three climatic zones using logistic regression and the magnitude of association was expressed in terms of odds ratio and corresponding 95% confidence intervals. All the statistical analyses were carried out statistical software Stata 15.0 and R 3.4.4 while geographical maps were prepared using Esri ArcGIS 10.8 software.

## Results

Based on the Koppen climate classification, Indian districts (N = 692) may be divided into five (Arid, Cold, Polar, Temperate, and Tropical) major climate types. The majority of districts fall in Temperate (270 /692 (39.0%)) and Tropical (260/692 (37.6%)) regions, followed by Arid (N = 140 (20.2%)), Polar (N = 13 (1.9%)) and Cold (9/692 (1.3%)) regions (Fig 1). Based on exploratory data analysis it was observed that the three regions: Arid, Polar, and Cold are similar in terms of malaria caseloads, and so they were clubbed into one and called Other (Arid, Polar, and Cold). There was a statistical difference in malaria API over three climate zones

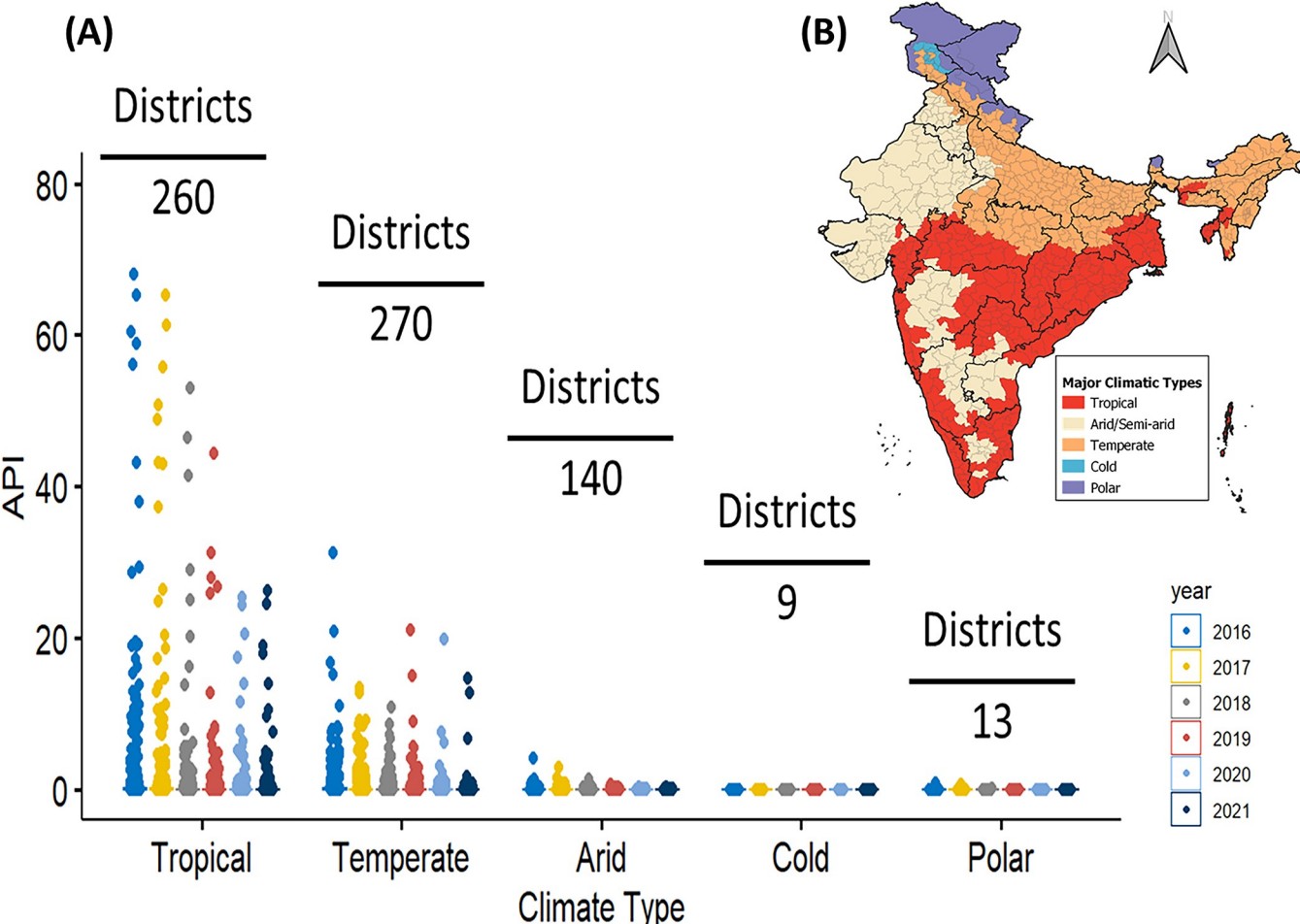

**Fig 1.** Distribution of malaria incidence (API) in (A) all major climatic zones in India from 2016 to 2021, (B) Geographical location of all major climatic zones.

**Table 1. Comparison of annual parasite incidence among three major regions of climate as per Koppen's classification.**

| Year | API | Koppen Climate Classification | | | P-value (Kruskal-Wallis Test) | Post-hoc Comparison (rank-sum test) | | |
|------|-----|-------------------------------|---|---|-------------------------------|-------------------------------------|---|---|
| | | Arid/Cold/Polar (G1) N = 162 | Temperate (G2) N = 270 | Tropical (G3) N = 260 | | G1 Vs G2 | G1 Vs G3 | G2 Vs G3 |
| 2016 | Mean ± SD | 0.14±0.36 | 1.21±3.56 | 3.64±11.24 | <0.001 | <0.001 | <0.001 | <0.001 |
| | P50 (P25 to P75) | 0.04(0.01 to 0.14) | 0.1(0.01 to 0.67) | 0.24(0.04 to 1.11) | | | | |
| | Min to Max | 0 to 4.07 | 0 to 31.24 | 0 to 88.47 | | | | |
| 2017 | Mean ± SD | 0.12±0.29 | 0.73±1.87 | 2.88±9.42 | <0.001 | <0.001 | <0.001 | 0.002 |
| | P50 (P25 to P75) | 0.03(0.01 to 0.13) | 0.09(0.01 to 0.45) | 0.15(0.03 to 0.75) | | | | |
| | Min to Max | 0 to 2.92 | 0 to 13.53 | 0 to 65.24 | | | | |
| 2018 | Mean ± SD | 0.07±0.15 | 0.44±1.25 | 1.39±5.87 | <0.001 | <0.001 | <0.001 | 0.046 |
| | P50 (P25 to P75) | 0.02(0 to 0.06) | 0.06(0.01 to 0.24) | 0.07(0.01 to 0.33) | | | | |
| | Min to Max | 0 to 1.42 | 0 to 10.86 | 0 to 53.08 | | | | |
| 2019 | Mean ± SD | 0.04±0.09 | 0.4±1.78 | 1.07±4.59 | <0.001 | <0.001 | <0.001 | 0.156 |
| | P50 (P25 to P75) | 0.01(0 to 0.04) | 0.03(0.01 to 0.17) | 0.04(0.01 to 0.16) | | | | |
| | Min to Max | 0 to 0.68 | 0 to 21.14 | 0 to 44.31 | | | | |
| 2020 | Mean ± SD | 0.02±0.04 | 0.2±1.37 | 0.75±3.07 | <0.001 | <0.001 | <0.001 | <0.001 |
| | P50 (P25 to P75) | 0(0 to 0.01) | 0.01(0 to 0.03) | 0.02(0 to 0.11) | | | | |
| | Min to Max | 0 to 0.28 | 0 to 19.83 | 0 to 25.32 | | | | |
| 2021 | Mean ± SD | 0.02±0.04 | 0.17±1.25 | 0.70±3.07 | <0.001 | <0.001 | <0.001 | <0.001 |
| | P50 (P25 to P75) | 0(0 to 0.01) | 0.01(0 to 0.02) | 0.01(0 to 0.08) | | | | |
| | Min to Max | 0 to 0.3 | 0 to 14.76 | 0 to 26.22 | | | | |

when compared on the year 2016's malaria data. Districts belonging to tropical (P50 (P25 to P75): 0.24(0.04 to 1.11); Min-Max: 0 to 88.47) and temperate regions (P50 (P25 to P75): 0.1 (0.01 to 0.67); Min-Max: 0 to 31.24) had higher API as compared to other regions (Arid/Cold/Polar) (P50 (P25 to P75): 0.04(0.01 to 0.14); Min-Max: 0 to 4.07). At the same time, there was a statistical difference between tropical and temperate regions as well. Similar conclusions can also be drawn when these comparisons were made on 2017 to 2021 data sets. All three climate regions have had statistically significant APIs over the years. Though there has been a significant reduction in malaria API over the years but the difference in tropical, temperate, and other climate zone remained statistically incomparable (Table 1).

To further assess this association, districts' API was classified into two categories: <1 and ≥1, and the distribution of districts with ≥1 API was compared over these three climate zones, and the magnitude of association was assessed in terms of the odds ratio. The odds of having API>1 was 21.9 (95% CI: 5.7 to 60.0) and 28.3 (95% CI: 7.7 to 80.2) times higher in the temperate and tropical zones respectively as compared to other zones (Arid/Cold/Polar) in 2016 (Table 2). Similarly, the odds ratio for API >1 in temperate and tropical regions in the years 2017 (14.7 and 22.5) and 2018 (17.9 and 23.4) were found to be high and statistically significant as compared to the other regions (Arid/Cold/Polar). Odds ratio calculation for the years 2019, 2020, and 2021 could not be performed as none of the districts had an API>1 in other regions (Arid/Cold/Polar) which were used as a reference category in the calculation of odds ratio estimation. Nevertheless, more than 5% and 11% of the total number of districts in temperate and tropical zones respectively had API of more than 1 as compared to 0% of districts in other regions in 2019. Data from 2020 and 2021 also depict the same phenomenon (Table 2).

The distribution of high-burden malaria districts (API ≥ 1) of India was mapped under three climate zones for the year 2016 to 2021 separately. This indicated that a large number of

**Table 2. Association of different climatic zones as per Koppen classification with high malaria endemicity over the years.**

| Year | Koppen Climate Classification | Districts with | | Odds Ratio (95% CI) | P value |
|---|---|---|---|---|---|
| | | API<1 | API≥1 | | |
| 2016 | | N = 564 | N = 128 | | |
| | Arid/Cold/Polar | 160 (28.37%) | 2 (1.56%) | Ref | |
| | Temperate | 212 (37.59%) | 58 (45.31%) | 21.89 (5.27-90.95) | <.0001 |
| | Tropical | 192 (34.04%) | 68 (53.13%) | 28.33 (6.84-117.42) | <.0001 |
| 2017 | | N = 591 | N = 101 | | |
| | Arid/Cold/Polar | 160 (27.07%) | 2 (1.98%) | Ref | |
| | Temperate | 228 (38.58%) | 42 (41.58%) | 14.74 (3.52-61.76) | <.0001 |
| | Tropical | 203 (34.35%) | 57 (56.44%) | 22.46 (5.40-93.41) | <.0001 |
| 2018 | | N = 631 | N = 61 | | |
| | Arid/Cold/Polar | 161 (25.52%) | 1 (1.64%) | Ref | |
| | Temperate | 243 (38.51%) | 27 (44.26%) | 17.89 (2.41-132.96) | 0.005 |
| | Tropical | 227 (35.97%) | 33 (54.10%) | 23.41 (3.17-172.9) | 0.002 |
| 2019 | | N = 650 | N = 42 | NA | <0.001[*] |
| | Arid/Cold/Polar | 162 (24.9%) | 0 | | |
| | Temperate | 256 (39.38%) | 14 (33.33%) | | |
| | Tropical | 232 (35.69%) | 66 (66.67%) | | |
| 2020 | | N = 659 | N = 32 | NA | <0.001[*] |
| | Arid/Cold/Polar | 162 (24.58%) | 0 | | |
| | Temperate | 260 (39.62%) | 8 (25.0%) | | |
| | Tropical | 236 (35.81%) | 24 (75.0%) | | |
| 2021 | | N = 667 | N = 25 | NA | <0.001[*] |
| | Arid/Cold/Polar | 162 (100%) | 0 | | |
| | Temperate | 266 (96.65%) | 4 (16.0%) | | |
| | Tropical | 239 (90.77%) | 21 (84.0%) | | |

**NA**-Not applicable as there is no event in the reference category in such a situation odds ratio cannot be calculated; * p-value from chi-square test.

high malaria endemic districts (API>1) fall under temperate and tropical regions across all years. In 2016, there were ~ 128 districts with API≥1, and among these 68 (i.e. ~53%) belonged to the Tropical region, 58 (i.e. ~45%) fell in the Temperate region only 2 were in Other (i.e. ~1.5%). In 2018, there were 101 districts with API>1, and among these 57 and 42 were from temperate and tropical regions respectively and 2 were from Other. In 2018, there were only 61 districts left with API>1, out of which 33 and 27 were from temperate, tropical and only 1 was from Other. Similarly, there were ~42 (Tropical = 28, Temperate = 14, and Other = 1), 32 (Tropical = 24, Temperate = 8, and Other = 0), and 25 districts (Tropical = 21, Temperate = 4, and Other = 0) with API>1 in the years 2019, 2020 and 2021 respectively. All these districts were either tropical or temperate and none were in Other regions (Table 2 and Fig 2).

These data suggest that tropical and temperate zones have a significantly higher burden of malaria as compared to other zones, with tropical regions having the greatest burden. Since the release of the national strategic plan for malaria elimination [10], intervention efforts were significantly scaled up. Thus malaria burden in high endemic states such as Odisha, and in both tropical and temperate regions reduced significantly. Nevertheless, the overall burden of malaria was still higher in tropical regions in 2018. However, in 2019, this difference diminished and a comparable number of districts with high malaria burden were observed in both tropical and temperate regions. Since 2020, this difference has once again become significant (Table 2 and Fig 2).

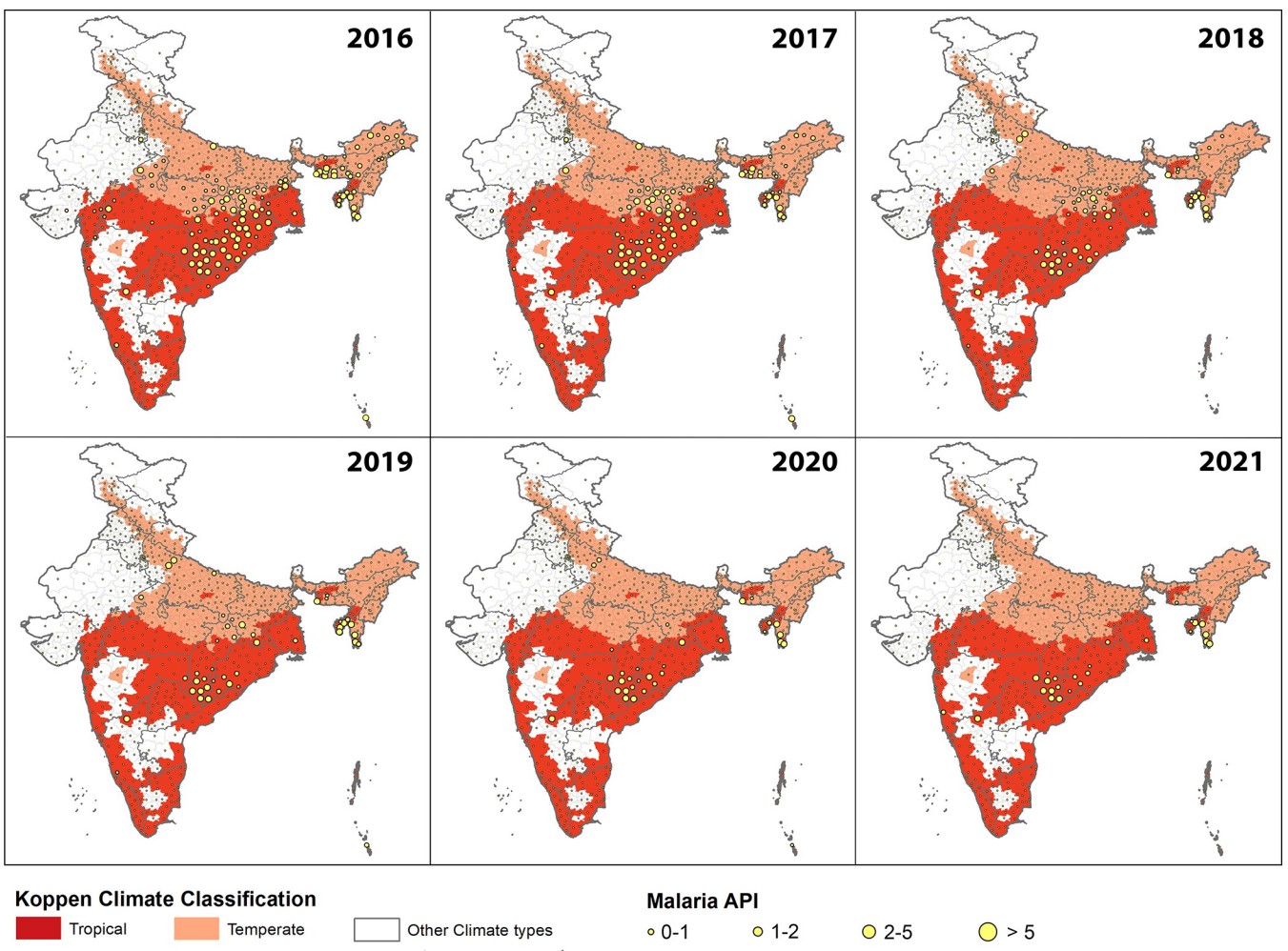

**Fig 2. Distribution of malaria incidence in tropical, temperate, and other climatic zones from 2016 to 2021 in India.**

When the share of malaria cases within each climatic sub-group was assessed, it was found that in tropical climates, the climate types Am (Tropical Monsoon) and Aw (Tropical wet savannah) were most suitable for supporting malaria transmission, whereas, in temperate climatic regions, the hot summer Mediterranean (Csa), Humid sub-tropical (Cfa), Monsoon humid sub-tropical (Cwa) and Sub-tropical Highlands (Cwb) seemed to support transmission. In the tropical and temperate climatic zones, India lacks regions of tropical dry savannah (As), warm and cold summer Mediterranean (Csb & Csc), Cold sub-tropical (Cwc), temperate oceanic (Cfb) and sub-polar oceanic (Cfc) type of sub-climates (Tables B and C in S1 Text).

A comparison of the present and future distribution of climate sub-types in India reveals that by 2100 there may be a significant expansion in the tropical monsoon (Am) climate towards central and northern India, and of tropical wet savannah (Aw) climate in the northeast of India. This is expected to increase the risk of malaria transmission in these regions as Am and Aw-type climates were found to be most suitable for malaria transmission. Furthermore, in the far north, the polar climate in the Leh-Ladakh region is expected to become a cold steppe arid desert, under a high level of climate change (RCP 8.5) in 2100 (Fig 3).

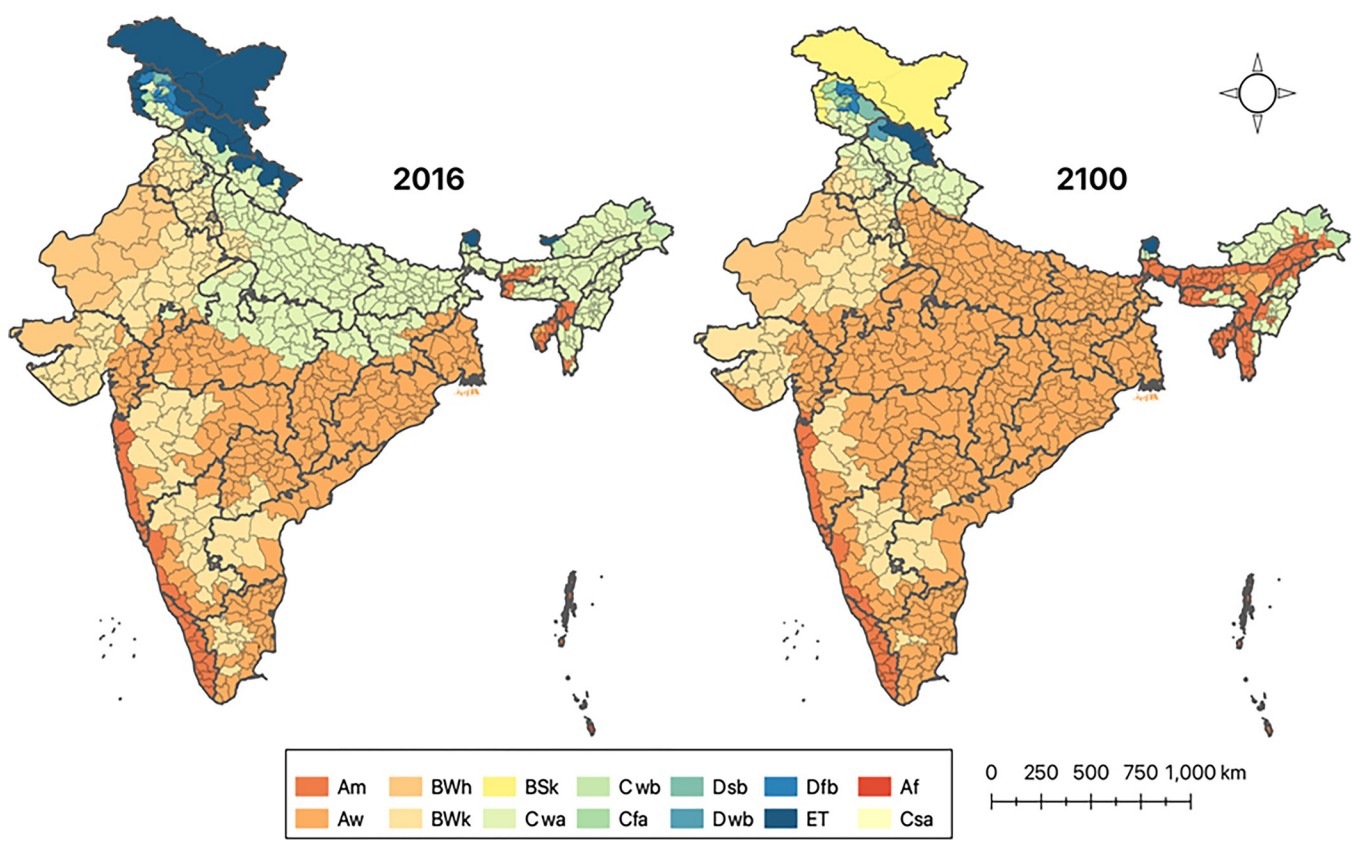

**Fig 3. Present (2016) and Future (2100) district-wise Koppen climate subclasses across India.**

## Discussion

Malaria is a persisting public health problem in India with ~90% of the country's population residing in malaria-endemic regions. This makes the control of malaria challenging, and after decades of efforts and interventions, we are once again on the path to elimination. To achieve this goal, it is crucial to efficiently allocate resources so that regions that are most at risk of malaria are better able to mitigate the risk and adapt to future changes. As malaria is a climate-sensitive vector-borne infection, its distribution is invariably linked to the climate type/sub-type of the region. Therefore, it is generally recognized that projected changes in temperature and rainfall possibly would affect or alter this distribution significantly. The present study was undertaken to understand malaria distribution across different climate types and sub-types in India, and to postulate the possible effects of climate change on this distribution across different climate types. As the study results suggest, the classification of districts based on different climate types and sub-types will help in understanding malaria transmission and will play a crucial role in identifying the regions which have the environmental potential for a future resurgence of malaria. The study provides loci where the control program needs to be extra vigilant as currently malaria may be under control but may rise if we lose our focus as the local climates will continue to support malaria transmission. Along with this, an environmental susceptibility indicator for malaria should be defined for all the districts of India and that should be used in the classification of districts into different categories for malaria elimination. Such reclassifications can be done digitally using the malaria dashboard (NIMR-MDB) where climate data can be overlaid [11].

Several earlier studies have employed climatic and ecological based factors to group regions based on homogeneity in malaria transmission and/or vector distribution so that resources can be easily distributed to regions most at risk [12–15]. Such an eco-regional classification has important decision-making consequences as it enables the tailoring of intervention programs that are specific to each category. One such classification described five eco-regions of malaria vector homogeneity, namely the coastal, piedmont, savannah, interior lowland forest, and high valley regions–it then identified the type of vectors present in each of these ecoregions [12]. Another study classified the different malaria zones based on the interaction of temperature and humidity which act through the mosquito vector as well as the parasite to manifest in variable endemicity [13]. In this classification, malaria transmission was divided into tropical malaria which is endemic, subtropical malaria which is characterized by severe epidemics of malaria, temperate malaria which occurs seasonally and equatorial malaria where malaria epidemics generally supersede periods of drought [13,14]. This classification was done within the neotropics, and it divided malaria epidemiological zones in South America and the West Indies into northern para-equatorial, western equatorial, eastern equatorial, and southern sub-tropical zones. Later on, the northern sub-tropical zone was added to this classification to describe malaria transmission in Central America and Mexico [15].

This study has observed significant statistical differences in the distribution of malaria cases across different climatic zones of the country. Overall, from 2016 to 2021, the tropical districts accounted for > 73% of the malaria burden, while the temperate regions contributed ~ 25% of the total malaria burden. Other climatic zones accounted for <2% of the total malaria burden. Almost all the high malaria burden districts (API≥1) were in the tropical and temperate regions, with a greater share of high malaria burden districts in the tropical regions. Furthermore, the number of high malaria burden districts was relatively comparable in tropical (53.15%) and temperate (45.31%) climatic regions in 2016 (Table 2). However, over the years, the difference in the share of high malaria burden districts between the tropical and temperate zones has greatly increased, and by 2021 the tropical regions accounted for 75% of all the high malaria burden districts, whereas the temperate region had only 25% of the high malaria burden districts (Table 2). The likely cause for this is that malaria is more or less stable in tropical regions with relatively higher API, whereas malaria is usually unstable and epidemic in temperate regions with relatively lower API. The average API in the tropical region is more than twice the average API in temperate regions for almost all the years i.e., 2016–2021 (Table 1). As a result of lower APIs, intervention efforts over the years have been successful in bringing down the malaria API below 1 in a greater number of high-burden districts of the temperate regions as compared to the tropical regions.

The tropical climate in India encompasses most of the central Highlands, northern parts of the Deccan plateau, the western Ghats, the southern peninsular plateau as well as some parts of the Indian northeast. The eastern parts of the country that experience this climate (including the states of Odisha, Chhattisgarh, West Bengal, parts of Jharkhand, Assam, Meghalaya and Tripura) are the epicenters of malaria in India and account for a majority of the malaria burden. While the western Ghats also account for some burden of malaria, the southern peninsular plateau is largely devoid of malaria despite lying in the tropical zone. The temperate region includes most of the Indo-Gangetic plains, the lower Terai regions before the Himalayas as well as the northeast. Almost all of this region falls under the Cwa sub-class, which is most similar to tropical climates, and also accounts for a large proportion of the malaria burden. The third most prevalent type of climate in India i.e. Arid/Semi-arid includes much of the western region (Rajasthan, Gujarat, Punjab, Haryana) and the Deccan plateau. This region generally has lower APIs ranging from 0–0.5. Cold and Polar climate regions are only present in the Himalayan region (including Jammu & Kashmir, Arunachal Pradesh, Uttarakhand, Himachal

Pradesh, and Sikkim) and higher altitudes, and experience only a small share of the malaria burden.

Among the tropical districts, the malaria burden was fairly high in both the Am (Tropical Monsoon) and the Aw (Tropical Wet Savannah) climate sub-types, with the odds of high malaria burden (API> = 1) tilted towards the Aw climate between 2015 to 2017 but leaning towards the Aw climate sub-type from 2018 onwards. Moreover, by 2020, both the climatic subtypes have an almost equal share of high (API≥1) and low (API<1) malaria burden districts. Aw, type of climate has alternating wet and dry seasons, with an extended rainy monsoon, and contains some of the most hyper-endemic malaria regions in India, particularly in the states of Odisha and Chhattisgarh. The Tropical Monsoon (Am) climate is characterized by higher rainfall as compared to the Aw type of climate, which possibly results in the washing away of vector habitats along the western Ghats due to proximity to the coastline, resulting in low transmission of malaria. However, in the northeastern regions of India, districts with this climate have high endemicity for malaria. High-burden malaria was also significantly prevalent in the temperate regions of India and was particularly discernable in the Cwa (monsoon humid sub-tropical) climate subtype, which is most similar to the tropical climate types. Other climatic sub-groups found within the temperate zone in India, namely the Cfa (Humid subtropical), Csa (Hot summer Mediterranean), and Cwb (Subtropical highland) sub-types, had a significantly lower share of the malaria burden, with the odds of high malaria burden 1.6 times more in Cwa sub-type as compared to other climate sub-types in the temperate zone. However, the difference is not statistically significant, likely due to the very few districts that have API higher than 1.

By the year 2100, under the most severe climate change scenario (RCP 8.5), models predict a significant recession in the monsoon-influenced humid subtropical climate (Cwa), which is projected to be replaced by the tropical wet savanna (Aw) in the Indo-Gangetic Plains and the tropical monsoon climate (Am) in the northeast. As these climate types are highly endemic for malaria, the suitability for malaria transmission is therefore projected to increase in the Indo-Gangetic plains as well as in the northeastern regions of India. At the same time, the monsoon-influenced humid subtropical climate recedes further northward into the states of Himachal Pradesh and Uttarakhand, where they may support higher-intensity malaria epidemics. Furthermore, the polar climatic region of Leh and Ladakh is projected to turn into a cold desert climate, though transmission of malaria may still not be supported in the region.

The strong association between the climate type/sub-type and malaria incidence highlights the need for the inclusion of climate type in the assessment and stratification of malaria risk. This will assist the control program to group districts based on similar malaria risk profiles, and to organize malaria elimination strategies accordingly keeping in focus climate change. Analyzing environmental data regularly and correlating it with malaria data using digital tools in auto mode will be essential in this context. The impact of climate change on malaria must be tracked and understood and alerts generated [11].

## Conclusions

We have compared the malaria burden across different climatic zones of India. The climate is a dominant player in malaria transmission and its usage as a malariometric is required for the ongoing malaria elimination efforts. This study highlights a significant difference in the distribution of malaria incidence across the tropical, temperate, polar, and cold climate types, with the tropical and temperate zones accounting for the vast majority of malaria cases. Within these climate types, the most suitable climate sub-types for malaria are the tropical wet savannah (Aw) and the monsoon-influenced humid sub-tropical climate (Cwa). Worryingly, these

climate subtypes are expected to expand in the Indo-Gangetic plains and the lower Himalayan region respectively by 2100, leading to a threat of a resurgence of malaria or a higher risk for malaria in coming decades, if malaria is not eliminated by then. Our study thus suggests that in malaria-endemic countries, climate/incidence analysis should be routinely done so that all nations aiming for malaria elimination can be in synchrony in the context of climate change and its impact on malaria.

## Supporting information

**S1 Text.** Table A: Description and temperature/precipitation characteristics of the different Koppen-Geiger Climate classes and sub-classes. Table B: Malaria burden comparison between subtypes of temperate region. Table C: Malaria burden comparison between subtypes of tropical regions.
(DOCX)

## Acknowledgments

We are very thankful to Directorate of NCVBDC for providing data and ICMR-NIMR for all logistical support.

## Author Contributions

**Conceptualization:** Chander Prakash Yadav, Amit Sharma.

**Data curation:** Chander Prakash Yadav, Syed Shah Areeb Hussain, Rajit Mullick, Amit Sharma.

**Formal analysis:** Chander Prakash Yadav, Syed Shah Areeb Hussain, Manju Rahi, Amit Sharma.

**Investigation:** Manju Rahi, Amit Sharma.

**Methodology:** Chander Prakash Yadav, Syed Shah Areeb Hussain, Manju Rahi, Amit Sharma.

**Project administration:** Manju Rahi, Amit Sharma.

**Resources:** Amit Sharma.

**Software:** Chander Prakash Yadav, Syed Shah Areeb Hussain.

**Supervision:** Manju Rahi, Amit Sharma.

**Validation:** Rajit Mullick, Manju Rahi, Amit Sharma.

**Visualization:** Chander Prakash Yadav, Amit Sharma.

**Writing – original draft:** Chander Prakash Yadav, Syed Shah Areeb Hussain, Rajit Mullick.

**Writing – review & editing:** Manju Rahi, Amit Sharma.

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
