## [Decision Letter · Decision Letter 0]

20 Mar 2023

PGPH-D-23-00323

Climate zones are a key component of the heterogeneous presentation of malaria and should be added as a malariometric for the planning of malaria elimination

Dear Amit,

Thank you for submitting your manuscript to PLOS Global Public Health. After careful consideration, we feel that it has merit but does not fully meet PLOS Global Public Health’s publication criteria as it currently stands. Therefore, we invite you to submit a revised version of the manuscript that addresses the points raised during the review process.

We look forward to receiving your revised manuscript.

Kind regards,

Collins Otieno Asweto, PhD

Academic Editor

Journal Requirements:

Reviewers' comments:

Reviewer's Responses to Questions

**Comments to the Author**

1. Does this manuscript meet PLOS Global Public Health’s publication criteria? Is the manuscript technically sound, and do the data support the conclusions? The manuscript must describe methodologically and ethically rigorous research with conclusions that are appropriately drawn based on the data presented.

Reviewer #1: Yes

Reviewer #2: Yes

Reviewer #3: Yes

Reviewer #4: Partly

2. Has the statistical analysis been performed appropriately and rigorously?

Reviewer #1: Yes

Reviewer #2: Yes

Reviewer #3: Yes

Reviewer #4: Yes

3. Have the authors made all data underlying the findings in their manuscript fully available (please refer to the Data Availability Statement at the start of the manuscript PDF file)?

Reviewer #1: Yes

Reviewer #2: Yes

Reviewer #3: Yes

Reviewer #4: Yes

4. Is the manuscript presented in an intelligible fashion and written in standard English?

Reviewer #1: Yes

Reviewer #2: Yes

Reviewer #3: Yes

Reviewer #4: Yes

5. Review Comments to the Author

Reviewer #1: Dear Authors,

Thanks for your work on malaria. The topic of your study is relevant and timely within the realms of public health. The statistical techniques are adequate and conclusions are based on the data, though I have requested that you crystallize these conclusions more succinctly. The manuscript is presented in clear English language.

Here are my comments

1. Line 21: I am rather more comfortable with “Abstract” rather than “Summary”

2. L21 – 26: You did not state the study objectives. Kindly refer to author guidelines for abstract. The study objectives should form part of the “background”

3. Kindly review the abstract to be sure the word length does not exceed 300 (refer to author guidelines).

4. L40: I suggest the word “Discussion” or “Conclusions” in place of “interpretation”

5. L105: Classification of climate: please confirm… (including WorldClim V1 & V1…): Are these two different sources?

6. L108: I suggest you justify the preference for RCP 8.5. For an instance, with all the global efforts to contain climate change, some other author may settle for RCP 4.5

7. In your discussion, I expected to see a comparison with some other health systems (China, Uzbekistan, Paraguay, Algeria, Argentina, El Savador ) where malaria has been totally eradicated; showing the possible contributions of considerations of climatic classification to their success stories.

8. L273: eastern equatorial appeared twice.

9. Conclusion: It appears that your conclusions have been subsumed in the Discussion section. It will be helpful if a clear conclusion is set apart.

10. Recommendation: Do you have any specific policy intervention you would recommend in the light of your findings?

Reviewer #2: The manuscript is technically sound and innovative. The authors should make more clarity between the parasite (Plasmodium spp) and its vector (Mosquito) on the effects of climatic factors during introduction in the manuscript. The author should also elucidate the relationship between the vector and parasite in tern of transmissibility of malaria disease. These two genuine concepts would enable the audience appreciation of the study for significance in the elimination of malaria in any regions in the world. All comments raised in the manuscript should be addressed as they are vital for the quality of the manuscript. All tracked changes in the manuscript should be addressed as well.

In summary, the manuscript can be published if all the comments and suggestions were considered.

Reviewer #3: The authors have presented a logical and well elaborated comparison of malaria incidence based on different geographical zones of climate in India.

They describe 3 climatic zones and present results of malaria incidence in the zones over 5 years. They conducted statistical analyses and present the results clearly: there was statistical differences in malaria incidence between zones the highest being in the tropical zone.

The interplay between humidity, temperature and monsoon activity is elaborated.

The proposal to use climatic data for future allocation of resources for malaria elimination is presented in a scientifically sound manner by the authors.

I have a comments for clarification

In the introduction, the authors wrote the following sentence:

Malaria is a climate-sensitive disease and different climates and landscape features play a significant role in the propagation of malaria vectors thereby directly influencing malaria incidence.

However, the discussion and conclusion do not clearly elucidate how and what landscape features play a role in malaria incidence. Landscape is not a feature of the Koppen-Geiger Climate Classes so further explanation of how it impacts ,malaria incidence is required.

Reviewer #4: The effect of climate on malaria endemicity is well known, the authors must add a clear justification of the rationale for conducting this study and explain the reasons for the post hoc analysis.

There is need for clarity on study design, the type of study is not clearly defined under the methods section.

6. PLOS authors have the option to publish the peer review history of their article (what does this mean?). If published, this will include your full peer review and any attached files.

**Do you want your identity to be public for this peer review?** For information about this choice, including consent withdrawal, please see our Privacy Policy.

Reviewer #1: No

Reviewer #2: No

Reviewer #3: No

Reviewer #4: **Yes: **Gift Tafadzwa Chareka

---

## [Decision Letter · Decision Letter 1]

25 May 2023

Climate zones are a key component of the heterogeneous presentation of malaria and should be added as a malariometric for the planning of malaria elimination

PGPH-D-23-00323R1

Dear Amit,

We are pleased to inform you that your manuscript 'Climate zones are a key component of the heterogeneous presentation of malaria and should be added as a malariometric for the planning of malaria elimination' has been provisionally accepted for publication in PLOS Global Public Health.

Best regards,

Collins Otieno Asweto, PhD

Academic Editor

Reviewer Comments (if any, and for reference):

Reviewer's Responses to Questions

**Comments to the Author**

1. If the authors have adequately addressed your comments raised in a previous round of review and you feel that this manuscript is now acceptable for publication, you may indicate that here to bypass the “Comments to the Author” section, enter your conflict of interest statement in the “Confidential to Editor” section, and submit your "Accept" recommendation.

Reviewer #1: All comments have been addressed

Reviewer #3: All comments have been addressed

2. Does this manuscript meet PLOS Global Public Health’s publication criteria? Is the manuscript technically sound, and do the data support the conclusions? The manuscript must describe methodologically and ethically rigorous research with conclusions that are appropriately drawn based on the data presented.

Reviewer #1: Yes

Reviewer #3: Yes

3. Has the statistical analysis been performed appropriately and rigorously?

Reviewer #1: Yes

Reviewer #3: Yes

4. Have the authors made all data underlying the findings in their manuscript fully available (please refer to the Data Availability Statement at the start of the manuscript PDF file)?

Reviewer #1: Yes

Reviewer #3: Yes

5. Is the manuscript presented in an intelligible fashion and written in standard English?

Reviewer #1: Yes

Reviewer #3: Yes

6. Review Comments to the Author

Reviewer #1: Dear Author,

Thanks for your response by way of improvements to the initial manuscript . I am satisfied with the manuscript in its current form and recommend that it should be accepted for publication.

Reviewer #3: (No Response)

7. PLOS authors have the option to publish the peer review history of their article (what does this mean?). If published, this will include your full peer review and any attached files.

**Do you want your identity to be public for this peer review?** For information about this choice, including consent withdrawal, please see our Privacy Policy.

Reviewer #1: No

Reviewer #3: No
